# Cost-utility analysis of home blood pressure measurement for screening and diagnosis of hypertension through village health volunteer mechanism in Thailand

**Auttakiat Karnjanapiboonwong[1], Usa Chaikledkaew[1,2]\*, Thunyarat Anothaisintawee[1,3], Naiyana Praditsitthikorn[4], Charungthai Dejthevaporn[5], Ammarin Thakkinstian[1,3]**

**1** Mahidol University Health Technology Assessment (MUHTA)Graduate Program, Mahidol University, Bangkok, Thailand, **2** Social and Administrative Pharmacy Division, Department of Pharmacy, Faculty of Pharmacy, Mahidol University, Bangkok, Thailand, **3** Department of Clinical Epidemiology and Biostatistics, Faculty of medicine Ramathibodi Hospital, Mahidol University, Bangkok, Thailand, **4** Department of Disease Control, Ministry of Public Health, Nonthaburi, Thailand, **5** Division of Neurology, Department of Medicine, Faculty of Medicine Ramathibodi Hospital, Mahidol University, Bangkok, Thailand

\* usa.chi@mahidol.ac.th

## Abstract

This study aimed to evaluate the cost-effectiveness of blood pressure (BP) screening strategies, including 1) home blood pressure measurement (HBPM), (2) serial screening by CBPM followed by HBPM among individuals with high BP i.e., clinic BP $\geq$140/90 mmHg (Serial1), (3) serial screening by CBPM followed by HBPM among individuals without high BP i.e., clinic blood pressure <140/90 mmHg (Serial2) compared to CBPM alone. A Markov model was applied among Thai population aged 35 years who had not been previously diagnosed with hypertension (HT) during a lifetime horizon with one-year cycle length from a societal perspective. One-way and probabilistic sensitivity analyses using Monte Carlo simulation with 1,000 replications were performed. The total cost of Serial2 (118,283 baht) was the highest and followed by HBPM (110,767 baht), CBPM (110,588 baht) and Serial1 (78,310 baht). The total quality adjusted life years (QALYs) for the population undergoing BP screening with CBPM, HBPM, Serial1, and Serial2 were 22.1557, 22.1511, 22.1286, and 22.1564, respectively. Compared to CBPM, Serial1 was associated with an incremental cost saving of 32,278 and an incremental QALY loss of 0.0271, whereas HBPM was dominated by CBPM due to higher cost (179 baht) and fewer QALY (-0.0046). Additionally, the incremental cost-effectiveness ratio (ICER) of Serial2 was the highest (10,992,000 baht per QALY gained). Moreover, the incidence rate of HT among individuals at age 40–49 years was the most sensitive factor influencing the ICER of HBPM, Serial1 and Serial2. At the Thai societal willingness-to-pay (WTP) threshold of 160,000 baht per QALY gained, the cost saving associated with Serial1 outweighed the QALY loss. Therefore, it is recommended that Serial1 be implemented as a BP screening option in Thailand. This evidence informed policy information could be invaluable for policymakers in making decision regarding BP screening through village health volunteer mechanism in Thailand and similar settings.

**Data Availability Statement:** Data cannot be shared publicly because there are ethical restrictions on publicly sharing a data set. Data are available from the Research Ethics Protection Unit, the Faculty of Dentistry/Faculty of Pharmacy, Mahidol University (contact via Dental Simulation and Research Building, 5th Floor Mahidol University Faculty of Dentistry, 6, Yothi Road, Ratchathewi, Bangkok, 10400 or +662-200-7622) for researchers who meet the criteria for access to confidential data.

**Funding:** This work was supported by funding from Mahidol University and the International Decision Support Initiative (iDSI) through the doctoral study in Mahidol University Health Technology Assessment (MUHTA) Graduate Program. This work was produced as part of the iDSI (www.idsihealth.org), which supports countries to get the best value for money from health spending. iDSI receives funding support from the Bill & Melinda Gates Foundation and the UK Department for International Development (Grant No. OPP1087363). The funders had no role in the study design, data collection and analysis, decision to publish, or preparation of the manuscript.

**Competing interests:** The authors have declared that no competing interests exist.

# Introduction

Hypertension (HT) is a long-term) medical condition in which the blood pressure (BP) in the arteries is persistently elevated [1]. The presence of HT can substantially elevate the risk of various cardiovascular diseases (CVD), including coronary heart diseases (CHD), congestive heart failure (CHF) and peripheral arterial diseases [2]. Moreover, it is a major contributor to the incidence of both ischemic and hemorrhagic stroke, as well as renal failure [2]. The World Health Organization (WHO) rates HT as one of the most important causes of premature death worldwide, estimated to cause 7.5 million deaths or approximately 12.8% of the total of all deaths [3]. According to the 2014 National Health Examination Survey in Thailand, HT prevalence was about 24.7% and caused more than 50,000 deaths annually due to CVD including stroke and ischemic heart disease (IHD) [4]. However, only one third of HT patients have their BP under control owing to limited access to diagnosis and treatment, especially for the working population [4]. As a result, screening and diagnosis for HT are important strategies to identify high-risk groups for early detection and timely treatment, which can lead to a decrease in HT related to morbidity and mortality [5].

Currently, several measurements have been used for diagnosis in practice i.e., ambulatory blood pressure measurement (ABPM), home blood pressure measurement (HBPM), and clinic blood pressure measurement (CBPM) [6–10]. ABPM is a diagnostic test that automatically measures the presence of HT at repeated intervals during normal daily activities over 24 consecutive hours [11, 12]. It has been widely accepted as a standard test for HT diagnosis, but is not routinely applied in clinical practice due to its high price and low compliance [12–15]. HBPM is a technique in which individuals take self-blood pressure measurement outside of clinic settings, such as at home, workplace, and communities and is suggested to be an alternative of ABPM, as it is inexpensive and convenient [16]. CBPM is a BP measurement measured in a clinic setting and is more commonly used in clinical practice; however, it may lead to over-diagnosis, i.e., white coat hypertension (WCHT) and under-diagnosis, i.e., masked hypertension (MHT) [17–20]. According to existing international guidelines, either HBPM or ABPM is recommended to confirm diagnosis of HT [8, 10, 21].

According to a recent systematic review by Wang et al. 2003 [22], four studies compared HBPM to CBPM. Two of these studies focused on HT management [23, 24], while the other two were for diagnosis purpose [25, 26] However, the latter two studies were cost analysis studies that demonstrated medical cost savings in screening with HBPM. Furthermore, the cost-effectiveness study by Lovibond et al. compared screening with ABPM and HBPM to CBPM. The study concluded that screening with ABPM was the most cost-effective strategy across all age groups (50, 60, 70, and 75 years) [27]. Additionally, in younger age groups, ABPM yielded greater cost savings but was associated with a slight reduction in quality adjusted life years (QALYs); nevertheless, it remained the most cost-effective option [27].

Although ABPM is considered the gold standard, its device is still expensive and may not be applicable in Thai clinical settings [21]. Currently, the national BP screening policy targets individuals aged over 35 years, who should undergo annual screening by village health volunteers (VHV) using CBPM. In addition, Thailand has promoted serial screening, which involved HBPM among individuals who receive CBPM screening and have clinic BP readings of 140–179 mmHg for systolic blood pressure (SBP) and/or 90–109 mmHg for diastolic blood pressure (DBP) [28]. However, no cost-effectiveness study of BP screening strategies has been conducted. Recently, according to Thai Health Technology Assessment (HTA) Guidelines, the societal willingness to pay (WTP) threshold in Thailand has been set at 160,000 baht per QALY gained [29, 30]. This implies that a BP screening strategy with an incremental cost-effectiveness ratio (ICER) below this threshold is considered a cost-effective option and may

be considered for inclusion as a national BP screening policy. Therefore, the objective of this study was to assess the cost-effectiveness of BP screening strategies in Thailand. The findings from this study could serve as evidence informed policy information for policymakers in determining which BP screening strategy should be implemented in Thailand.

## Materials and methods

### Study design

We conducted a cost-utility analysis using a Markov model to compare costs and health outcomes of BP screening strategies during lifetime horizon with one-year cycle length. Health outcomes were life years (LYs) and QALYs, the multiplication of LYs and utility score. The analysis was performed based on a societal perspective, considering both future costs and health outcomes, which were discounted at a rate of 3% per year as recommended by the Thai HTA guidelines [31].

### Target population

In the base case scenario, the target population consisted of a hypothetical cohort of Thai adults aged 35 years who had never been diagnosed with HT and had no history of CVD i.e., CAD and cerebrovascular diseases.

### Ethics approval

The ethical approval of this study was granted by the Faculty of Dentistry/Faculty of Pharmacy, Mahidol University. The written informed consent was obtained from the VHV and local officers who participated in the interviews for data collection on costs.

### Intervention and comparator

Studied interventions included (1) HBPM i.e., individuals measuring their BP twice a day at home for seven consecutive days, (2) serial screening by CBPM followed by HBPM among individuals with high BP i.e., clinic BP ≥140/90 mmHg (Serial1), (3) serial screening by CBPM followed by HBPM among individuals without high BP i.e., clinic blood pressure <140/90 mmHg (Serial2) and (4) CBPM as a comparator.

### Model structure

Fig 1 illustrates a Markov model used in this study. The model consisted of nine health states, i.e., (1) No HT, (2) Undiagnosed HT, (3) New HT diagnosis, (4) WCHT (temporary state), (5) WCHT (transient stage), (6) HT controlled, (7) HT uncontrolled, (8) CVD, and (9) Death. Initially, Thai adults aged 35 years with no prior diagnosis of HT or no history of CVD would enter to "No HT" health state. Based on the diagnostic performance of each screening test, some adults in the "No HT" state would move to "New HT diagnosis", "Undiagnosed HT" or "WCHT (temporary stage)". For the "WCHT" state, individuals would receive a diagnosis of HT despite not having HT prior to screening. New cases of WCHT (temporary stage) would temporarily remain in this state before moving to "WCHT (transient stage)" within the same cycle. Once diagnosed with HT, individuals would transition to the "HT uncontrolled" state, with the possibility of moving to "HT controlled" if their BP is successfully managed to <140/90 mmHg for SBP and DBP, respectively. In addition, cases in the "WCHT (transient stage)" cases would transition to "HT controlled". Individuals with undiagnosed HT, uncontrolled HT, or controlled HT had the potential to transition to "CVD" health state. All health states had the possibility of transitioning to the "Death" state and all health states except "New HT

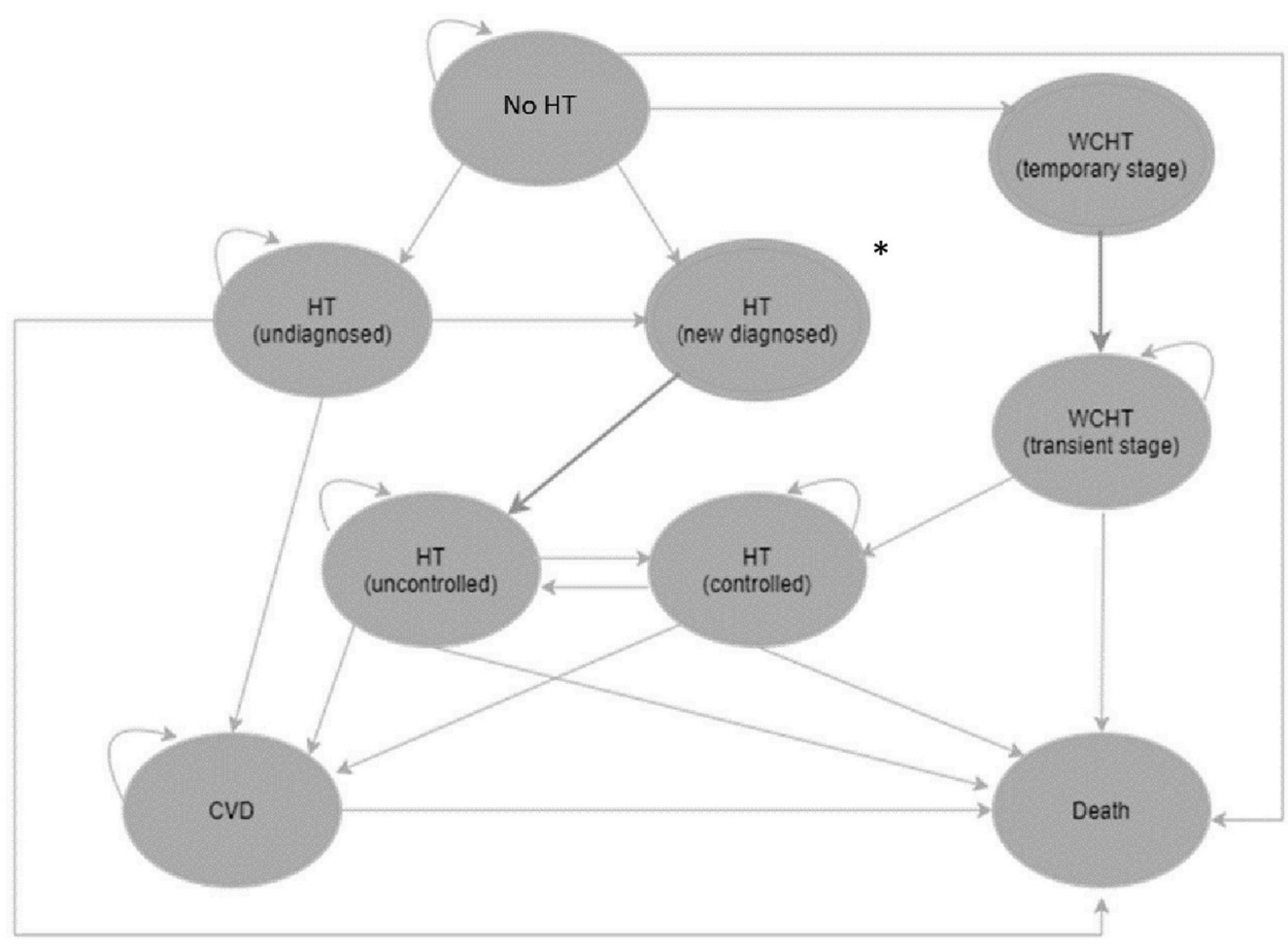

*Temporary stage

**Fig 1. Markov model.** CVD; cardiovascular disease, HT; hypertension, WCHT; white coat hypertension.

diagnosis" and WCHT (temporary state) would stay in the same health states. Moreover, adults with MHT were perceived as having normal BP by physicians and would have a chance to receive BP screening in the subsequent year. Similarly, individuals with HT and WCHT did not undergo BP screening again, as they were managed as HT cases. Individuals with MHT and high BP were classified within the undiagnosed HT group, with their CVD risk assumed to be equal to that of the uncontrolled HT group.

## Model input parameters

**Transitional probabilities.** S1 Table demonstrates a summary of the model input parameters used in this study (Supplementary Information). The HT incidence and prevalence data were obtained from the cohort of 2,235 officers aged 30–79 years employed by the Electricity Generating Authority of Thailand (EGAT) during two periods: 2012–2014 and 2017–2019.

The EGAT study represents one of the largest longitudinal cardiovascular cohort studies in the Thai population. Participants with a wide range of socio-demographic backgrounds located in Bangkok and three different sites in Western and Northern Thailand were randomly enrolled in this study. Subsequently, these data were adjusted using the fraction of non-age standardized HT prevalence from the 5th National Health Examination Survey (NHES 5th) which serves as the officially accepted 5-year round national survey for health statistics in Thailand [4].

In this model, CVD consisted of hemorrhagic stroke, ischemic stroke, and CAD. The prognosis of CVD varied depending on the specific disease and could be classified as acute or chronic. The probabilities of death due to hemorrhagic stroke, ischemic stroke, and CAD were borrowed from published literature reviews [32–35]. The overall annual probability of death due to CVD was computed by weighting the probabilities of death by the proportion of each disease in all CVD cases according to the Thailand Health Data Center (HDC) [36]. Additionally, the study of Chantaraprapabkun et al [35] revealed that among HT patients aged 15–45 years, the probability of death due to CVD was 0.00699, which was lower than those older than 45 years. Therefore, a fatal probability of 0.00699 was applied in the model for individuals younger than 45 years, while a probability of 0.0853 was used for those aged 45 years and older. In addition, the probability of death due to other causes was retrieved from Thailand Burden of Disease report [37].

The transitional probabilities from controlled HT to uncontrolled HT and vice versa were analyzed using data from HT patients diagnosed between 2014 and 2018. The probability of developing HT among individuals with WCHT was assumed to be equal to the incidence of HT among the general population. Moreover, evidence regarding the diagnostic performance of the screening test was retrieved from the meta-analysis conducted by Karnjanapiboonwong et al [38]. The diagnostic performance of serial screening was calculated based on sensitivity and specificity of CBPM and HBPM.

**Cost.** Based on a societal perspective used in this study, direct medical and non-medical costs were included. To avoid double counting in our cost-utility analysis, indirect costs such as lost income due to absenteeism at work or foregone wages while attending hospital or clinic appointments were excluded. However, we did incorporate the loss of income for caregivers, classifying it as direct non-medical costs in this study. Direct medical costs included cost of HT screening and treatment costs for stroke, CAD, and HT. Direct non-medical costs i.e., transportation, food, and caregivers' income were obtained from the Thai Standard Cost lists for Health Technology Assessment [38–43]. Cost of HT screening was estimated by gathering data on HT screening-related activities through interviews with health officers and VHV in Chachoengsao province during April-December 2019. The cost among VHV was estimated based on the duration and frequency of their roles, which included tasks such as preparing lists of targets, providing BP screening with instructions, ensuring coverage of BP screening through home visits, recording BP results, and transmitting the results to primary care units. The cost of health officers was determined by considering the frequency and workload of doctors and nurses involved in HT diagnosis in the extended outpatient department in the communities. Treatment cost for stroke, CAD, and HT were obtained from published literature reviews. Concerning transportation cost, it was assumed that HT patients with controlled BP would visit a doctor four days per year, while those with uncontrolled BP would visit eight days per year. For stroke cases, the initial hospital admission during the acute stage was estimated to last approximately 9.83 days based on the most recent evidence from research conducted under the Thai context [44]. Stroke patients were assumed to visit a doctor approximately eight times per year thereafter. All costs were adjusted to 2021 value using the consumer price index (CPI).

**Utility.** The utility scores were derived from published literature reviews. Utility values obtained from the EurolQoL-5D (EQ-5D) questionnaire of HT, stroke, and CAD were 0.95 [45], 0.55 [46], and 0.75 [31], respectively. The overall utility for CVD (0.62) was calculated using the utility of CAD and stroke weighted by the proportion of CAD and stroke cases in all CVD cases according to Thailand's HDC. The proportion of stroke cases in all CVD cases was 0.64, while the proportion of CAD in all CVD was 0.36 [47]. Therefore, the overall utility of CVD was calculated as (0.55 x 0.64) + (0.75 x 0.36) = 0.62. The utility of individuals without HT or those considered healthy as well as individuals with WCHT, was assumed to be 1, whereas the utility of death was assumed to be 0.

## Model validation and calibration

Face validity was conducted by a panel of experts including cardiologists, neurologists, internal medicine specialists, biostatisticians, health economists, and clinical epidemiologists and health technology assessment experts. Their roles were to assess and validate the model structure, input parameters and data sources used in this study. Moreover, the criterion validity involved comparing the HT prevalence estimated by the model among individuals aged 35–79 years with the actual prevalence reported in the 5th NHES in 2014 [4] (Fig 2).

The model aimed to replicate the actual population composition, including both patients and non-patients, by starting with a population that includes individuals with and without HT or CVD. This approach allowed for a comprehensive assessment of the model's performance in reflecting real-world scenarios. For model calibration, parameters such as age-specific incidence of HT in the Thai population and mortality rates from non-CVD cases were selected due to their association with high uncertainty. Following model calibration, HT prevalence within age group of 45–59, 60–69, and 70–79 years in the model was better fitted with those reported in the 5[th] NHES [4].

## Result presentation

Total costs, LYs, and QALYs for each BP screening option were presented over the lifetime horizon. The ICERs were calculated by diving the difference in costs divided by difference in LYs or QALYs, aiming to assess the cost-effectiveness of each BP screening option compared with CBPM. The Thai societal willingness to pay (WTP) threshold of 160,000 baht per QALY gained was used to determine cost-effectiveness.

## Statistical analysis

Statistical analysis was performed to evaluate the uncertainty surrounding the estimated ICERs and to determine the cost-effectiveness of the interventions using the Microsoft Excel 2019 (Microsoft, WA, USA). One-way sensitivity analysis was conducted to investigate the uncertainty associated with each parameter and the results were visually represented using Tornado diagrams. These diagrams provide a clear illustration of the impact of varying individual parameters on the analysis results. Additionally, probabilistic sensitivity analysis (PSA) was employed to estimate parameter uncertainties surrounding the ICER using Monte Carlo simulation, wherein parameters were simultaneously and randomly selected about 1000 times based on their probability distributions. The results were demonstrated as the cost-effectiveness planes and cost-effectiveness acceptability curves (CEACs). The cost-effectiveness plane consists of a scatter plot with the x-axis representing the difference in costs, and the y-axis depicting the difference in QALYs between studied interventions and comparator. Moreover, CEACs are a graphical representation to present the probability that an intervention is cost-effective across a range of WTP thresholds. At the Thai societal WTP of 160,000 baht per

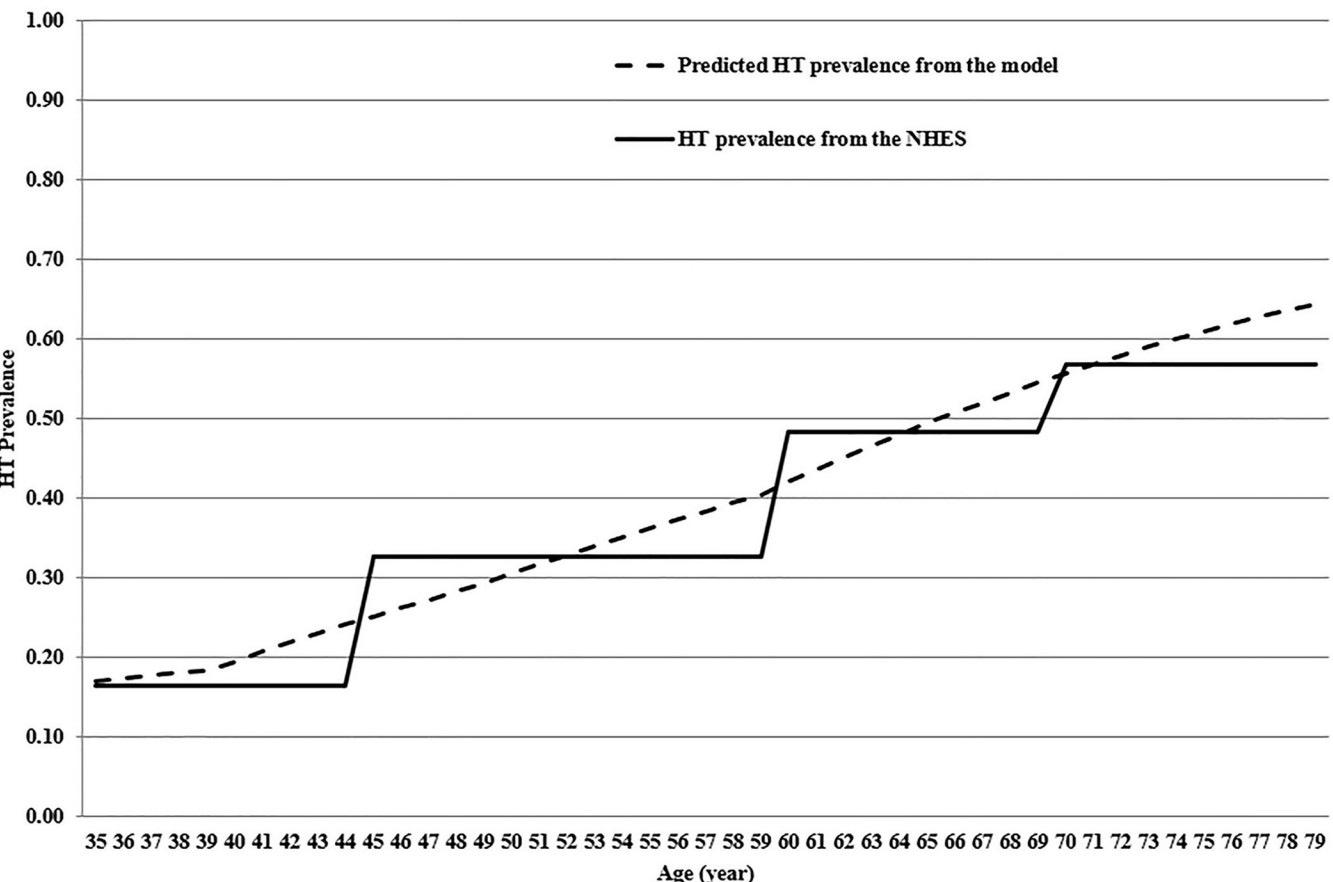

**Fig 2. Model validation.** HT; Hypertension.

QALY gained, the CEAC will indicate the probability that the intervention is cost-effective compared with the comparator.

## Results

Table 1 demonstrates costs and health outcomes. Compared to CBPM, Serial2 had higher HT prevalence and lower CVD mortality. In addition, Serial2 had lowest probability to be undiagnosed HT, but highest WCHT incidence. Based on a societal perspective, screening cost of CBPM was the lowest (76 baht), while that of HBPM was the highest (197 baht). Direct medical costs comprised the largest proportion of the total costs, accounting for approximately 60% for all BP screening strategies. The total cost of Serial2 (118,283 baht) was the highest and followed by HBPM (110,767 baht), CBPM (110,588 baht) and Serial1 (78,310 baht). The total QALYs of population with BP screening of CBPM, HBPM, Serial1, and Serial2 modality was 22.1557, 22.1511, 22.1286, and 22.1564, respectively. When compared with CBPM, Serial1 resulted in an incremental cost saving of 32,278 and an incremental QALY loss of 0.0271. However, HBPM was dominated by CBPM due to its high cost (179 baht) and lower QALY (-0.0046). In addition, Serial2 indicated higher costs (7,695 baht) and a slightly increase QALY (0.0007), resulting in the highest ICER value of 1,191,070 baht per QALY gained.

**Table 1. Cost and health outcomes.**

| Blood pressure screening strategy | Mean values | | | | Incremental costs and outcomes | | | ICER | | |
|---|---|---|---|---|---|---|---|---|---|---|
| | CBPM | HBPM | [a]Serial1 | [b]Serial2 | HBPM vs CBPM | [a]Serial1 vs CBPM | [b]Serial2 vs CBPM | HBPM vs CBPM | [a]Serial1 vs CBPM | [b]Serial2 vs CBPM |
| **Cost (baht)** | | | | | | | | | | |
| Screening cost (%) | 76 (0.07) | 530 (0.48) | 78 (0.1) | 197 (0.16) | 454 | 2 | 121 | | | |
| Treatment cost (%) | | | | | | | | | | |
| Direct medical cost | 66,891 (60.48) | 67,289 (60.75) | 46,015 (58.76) | 72,173 (61.02) | -673 | -11,405 | 2,301 | | | |
| Direct non-medical cost | 43,621 (39.45) | 42,948 (38.77) | 32,216 (41.14) | 45,922 (38.82) | 398 | -20,876 | 5,282 | | | |
| Total costs | 110,588 | 110,767 | 78,310 | 118,283 | 179 | -32,278 | 7,695 | | | |
| **Health outcomes** | | | | | | | | | | |
| HT prevalence (%) | 36.21 | 36.21 | 36.19 | 36.22 | 0 | -0.02 | 0.01 | | | |
| Incidence of CVD (%) | 19.78 | 19.79 | 19.97 | 19.78 | 0.01 | 0.19 | 0 | | | |
| CVD mortality (%) | 13.28 | 13.28 | 13.43 | 13.26 | 0 | 0.15 | -0.02 | | | |
| WCHT incidence | 0.763 | 0.748 | 0.426 | 0.798 | -0.015 | -0.337 | 0.035 | | | |
| Probability to be undiagnosed HT | 0.021 | 0.028 | 0.155 | 0.007 | 0.007 | 0.134 | -0.014 | | | |
| Total LYs | 39.9470 | 39.9408 | 39.9173 | 39.9443 | -0.0063 | -0.0298 | -0.0027 | [c]-28,570 | [d]1,083,896 | [c]-2,832,379 |
| Total QALYs | 22.1557 | 22.1511 | 22.1286 | 22.1564 | - 0.0046 | -0.0271 | 0.0007 | [c]-38,913 | [d]1,191,070 | 10,992,857 |

HT; hypertension, CVD; cardiovascular diseases, WCHT; white coat hypertension, LY; life years, QALY; quality adjusted life years, ICER; incremental cost-effectiveness ratio

[a]Serial1; CBPM followed by HBPM among people with clinic blood pressure ≥140/90 mmHg

[b]Serial2; CBPM followed by HBPM among people with clinic blood pressure <140/90 mmHg

[c]Negative ICER due to higher costs but less LYs or QALYs compared to CBPM

[d] Positive ICER due to less cost and less LYs or QALYs represents cost-saved per QALY lost.

## Uncertainty analysis

The results of one-way sensitivity analysis revealed that the incidence rate of HT among individuals aged 40–49 years was the most sensitive factor affecting the ICER of HBPM, Serial1 and Serial2. The ICER of Serial2 was most sensitive to the sensitivity of CBPM, the incidence rate of uncontrolled HT leading to CVD among individuals aged 40–49 years, and the incidence rate of HT among individuals aged 50–59 years from a societal perspective (Fig 3). To address the effect of sampling uncertainty on estimated incremental cost and incremental effectiveness parameters using Monte Carlo simulation in our single study-based economic evaluation, a cost-effectiveness plane demonstrated that most simulations of HBPM had higher costs and lower QALY, while Serial1 had substantially lower costs and less QALYs (Fig 4). In contrast, Serial2 was associated with higher costs and higher QALY. Additionally, the cost-effectiveness acceptability curves indicated that Serial1 had a 100% probability of being cost-effective compared with other BP screening modalities at the Thai societal WTP threshold of 160,000 baht per QALY gained (Fig 5).

## Discussion

This study represents the first attempt to conduct economic evaluation of the VHV-based BP screening modalities, including HBPM, Serial1 (initial screening by CBPM followed by HBPM among individuals with clinic blood pressure >140/90 mmHg), and Serial2 (initial screening

**A) HBPM compared with CBPM**

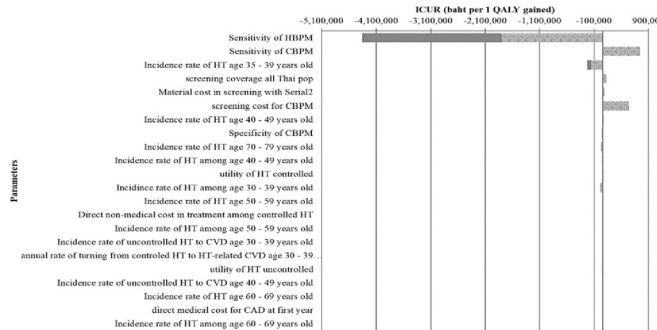

**B) Serial1 compared with CBPM**

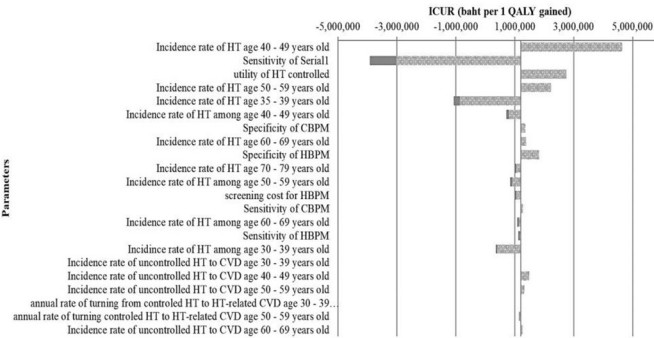

**C) Serial2 compared with CBPM**

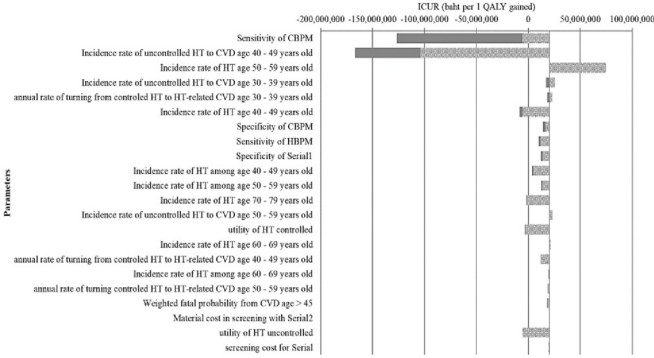

**Fig 3. Tornado diagram.** CBPM; clinic blood pressure measurement, HBPM; home blood pressure measurement, Serial1; CBPM followed by HBPM among people with clinic blood pressure ≥140/90 mmHg, Serial2; CBPM followed by HBPM among people with clinic blood pressure <140/90 mmHg.

by CBPM followed by HBPM among individuals with clinic blood pressure <140/90 mmHg) compared with CBPM, a current practice in Thailand.

Our results indicated that Serial1 was associated with an incremental cost saving of 32,278 baht and an incremental QALY loss of 0.0298. It is noteworthy that since Serial1 resulted in both negative cost and QALY increments, the ICER result could not be interpreted conventionally as the cost per additional QALY gained, but rather as the cost per QALY lost.

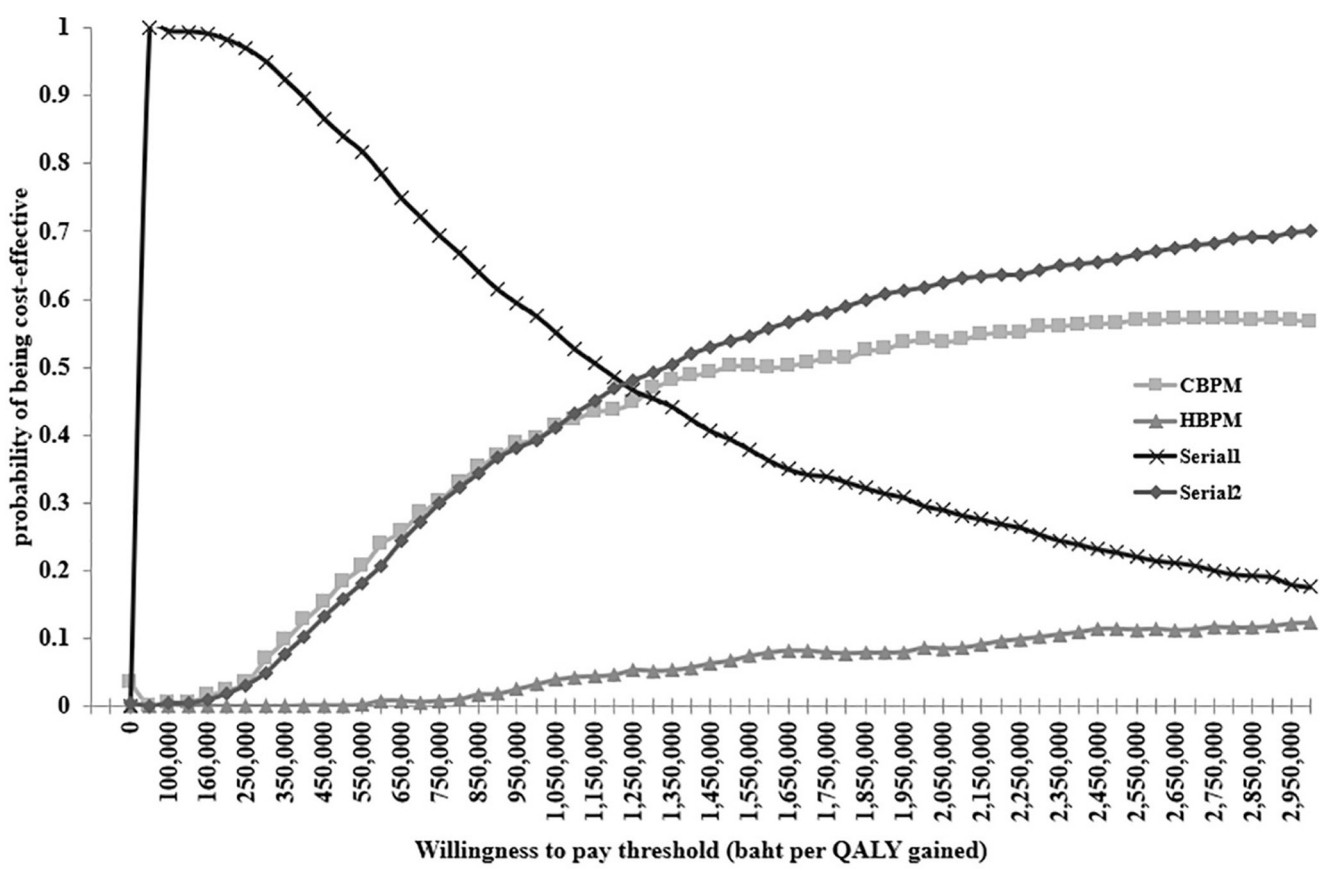

**Fig 4. Cost-effectiveness plane.** CBPM; clinic blood pressure measurement, Serial1; CBPM followed by HBPM among people with clinic blood pressure ≥140/90 mmHg, Serial2; CBPM followed by HBPM among people with clinic blood pressure <140/90 mmHg, QALY; quality adjusted life year.

Consequently, Serial1 could be expected to save 1,191,070 per QALY lost. At the Thai societal WTP threshold of 160,000 baht per QALY gained, Serial1 would be a cost-effective BP screening option when compared to CBPM, while HBPM and Serial2 would not be considered cost-effective over a lifetime period.

Furthermore, our findings also demonstrated that HBPM (-0.0046) and Serial1 screening strategies (-0.0271) resulted in lower QALYs, whereas the Serial2 screening strategy (0.0007) lead to slightly higher QALYs compared with CBPM. Interestingly, despite Serial2 yielding the highest QALYs and lowest CVD mortality, it was not deemed a cost-effective strategy due to its higher costs and minimal additional QALY gains. This finding was attributed to the superior diagnostic performance of Serial2 characterized by its highest sensitivity and lowest specificity, leading to a lower incidence of MHT and a higher incidence of WCHT compared to CBPM. Consequently, Serial2 effectively avoided MHT, which could potentially lead to harmful outcomes such as uncontrolled HT, thus resulting in the highest QALYs. In addition, compared to CBPM, both HBPM and Serial1 screening strategies led to fewer QALYs. This could be explained by the fact that Serial1 yielded the highest CVD incidence and mortality, while HBPM was associated with a higher incidence of CVD due to a higher incidence of undiagnosed HT, consequently resulting in lower QALYs.

Moreover, our study aligned with the findings of Lovibond et al., indicating that both the costs and QALYs of HBPM were similar to those of CBPM across all ages and genders [27].

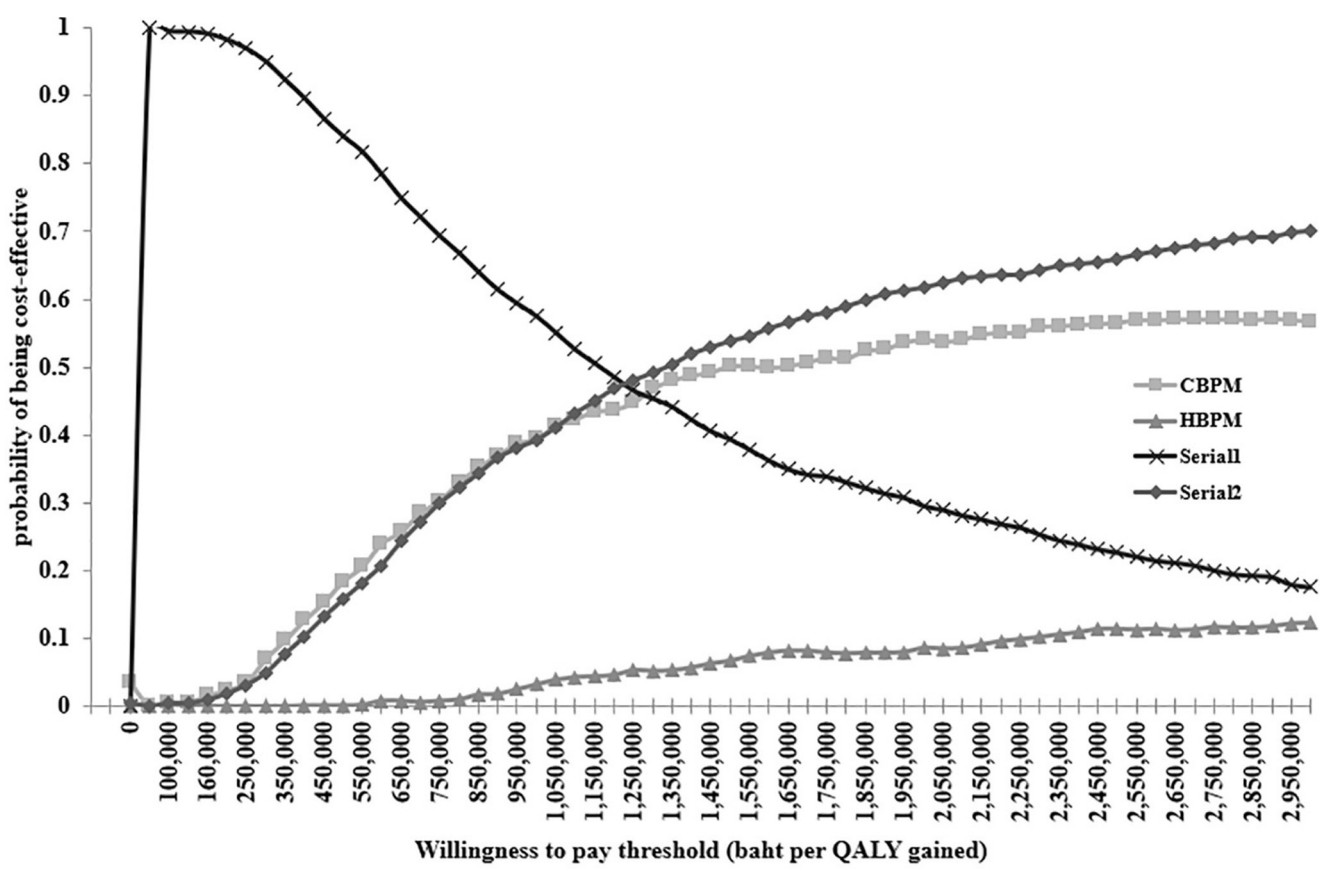

**Fig 5. Cost-effectiveness acceptability curve.** CBPM; clinic blood pressure measurement, Serial1; CBPM followed by HBPM among people with clinic blood pressure ≥140/90 mmHg, Serial2; CBPM followed by HBPM among people with clinic blood pressure <140/90 mmHg, QALY; quality adjusted life year.

However, our study differed from previously published studies that supported HBPM as a cost-saving strategy [47, 48]. For example, Fukunaga et al. demonstrated that HBPM could reduce medical costs by $US 1.56 million for every 1,000 patients over 5 years [25], particularly in scenarios with a high prevalence of WCHT, but low incidence of newly confirmed HT. Similarly, Funahashi et al. suggested that widespread implementation of HBPM could result in a substantial savings of $9.3 billion US dollars in HT-related medical costs mainly attributable to the identification and management of WCHT [26].

In terms of total costs, it was noteworthy to mention that more than 99% of total costs in this study were attributed to treatment costs, encompassing both direct medical and non-medical costs. Although the screening cost of Serial1 was almost the same as that of CBPM, it incurred significantly lower treatment costs compared to CBPM, HBPM, and Serial2. Specifically, the savings in direct medical and non-medical costs accounted for 20,876 and 11,405 baht, respectively. These significant cost reductions emphasized the potential cost-saving benefits of Serial1 in comparison to CBPM. Moreover, the Serial1 option yielded the lowest WCHT incidence, thereby potentially reducing unnecessary cost associated with WCHT, which represented a major part of treatment costs. The ability of Serial1 to correctly diagnose WCHT made it the preferred strategy for BP screening, especially in younger adults with a lower prevalence of HT, as recommended by the 2019 Thai Guideline for HT management [10].

Although MHT may theoretically lead to more clinically harmful, regular annual BP screening can reduce its prevalence, resulting in a lower incidence and mortality of CVD. Thus, the implementation of Serial1 as a screening modality can potentially mitigate the burden of HT related complications and associated healthcare costs.

The results of this study aligned with the recommendations set forth by the Thai Guidelines for Hypertension Management 2019, which advocated for the utilization of HBPM as a serial screening method subsequent to CBPM [10]. This sequential approach aims to reduce the incidence of false positive HT or WCHT [10]. However, the guidelines have recently expanded the use of HBPM to include individuals who have undergone CBPM and exhibited clinical BP values between 130/85 and <140/90 mmHg. This broadened application of HBPM aims to enhance the detection of HT among individuals with elevated but not yet hypertensive clinical BP levels. Despite these recommendations, our study findings revealed that Serial2 yielded cost-ineffective results. In Thai current clinical practice, only individuals with suspected HT or clinical BP exceeding 140/90 mmHg by CBPM commonly receive additional HBPM for diagnosis, highlighting a potential discrepancy between guideline recommendations and real-world implementation.

It was important to emphasize that our study results recommended the implementation of Serial1 which involved HBPM for individuals with clinic BP reading exceeding 140/90 mmHg by CBPM. HBPM requires individuals measuring their BP twice a day at home, typically in the morning and evening, over the course of seven consecutive days. While various cross-sectional studies have evaluated the optimal frequency of HBPM in specific cohorts of HT patients, there may be practical to reduce the duration of HBPM [48–52]. For instance, two studies by Stergiou et al. demonstrated that at least 12 measurements taken over three days were superior to conventional measurements [50, 51]. Future study should further investigate the optimal number of HBPM required in the Thai context. Nevertheless, it is crucial to consider various aspects such as quality of life and the indirect impact of high-cost healthcare payment to the health system when making decisions regarding population health benefits.

It was significant to highlight the limitations of our study. Firstly, the definition of WCHT was basically used when false positive HT occurred from measurement at doctor office. However, our study applied the term of WCHT for false positive HT occurred from either CBPM or HBPM. This broad definition might have led to the overestimation of WCHT in our study. Secondly, we assumed full adherence to the screening protocol, which might not reflect real-world practice. Studies have shown varying levels of adherence to HBPM schedules, with only a fraction of individuals achieving full adherence. For example, only 23.6%, 47%, and 89% had full adherence to a 28-day [53], 7-day [54], and 4-day HBPM schedules [54]. This could potentially lead to the overestimation of our results. Thirdly, while prolonged periods of BP measurement may lead to a slight improvement in diagnostic accuracy [52], our study did not fully explore this aspect. However, all these issues likely had minimal impact on the average BP values [53]. Fourthly, the cost data for screening activities were collected from two sub-districts in Cha-Choeng-Sao province, a rural and sub-urban area, which might not be representative of all provinces in Thailand. This limitation might restrict the generalizability of our findings to the entire Thai population. Lastly, our study utilized data from the EGAT cohort, which is one of the largest longitudinal cardiovascular cohort studies in the Thai population. The HT incidence estimated from this cohort might not perfectly represent the entire country, as the socio-economic status of EGAT employees differed from that of many economically disadvantaged individuals in Thailand. Additionally, the 'healthy worker effect' might influence baseline BP profiles from the EGAT cohort, further limiting the generalizability of our results [55]. Nevertheless, the data from EGAT cohort provide the best available evidence for Thai population with HT.

## Conclusions

When compared with CBPM within the context of a VHV-based BP screening campaign in Thailand for a population aged 35 years, our study revealed that Serial1 yielded an incremental cost saving of 32,278 baht and an incremental QALY loss of 0.0271. At the societal willingness-to-pay (WTP) threshold of 160,000 baht per QALY gained, the cost saving associated with Serial1 outweighed the QALY loss. On the other hand, Serial2 was not deemed a cost-effective option due to its much higher costs and minimal gains in QALYs. Moreover, HBPM was dominated by CBPM due to its higher costs and lower QALYs. Therefore, based on our findings, it is recommended that Serial1 be implemented as the preferred BP screening option in Thailand. This evidence-informed policy information could be invaluable for policymakers in making rational resource allocation decisions for BP screening through the VHV mechanism in Thailand and similar settings.

## Supporting information

**S1 Table. Parameters used in the model after calibration.**
(DOCX)

**S1 Checklist. CHEERS 2022 checklist.**
(PDF)

**S1 File. Interview questions to estimate the cost of screening.**
(DOCX)

## Acknowledgments

We would like to thank the Division of Non-communicable diseases, Department of Disease Control for all guidance and Bang Klue sub-district health office, and Bang-Pra-Kong sub-district health office for providing data in this study.

## Author Contributions

**Conceptualization:** Usa Chaikledkaew, Thunyarat Anothaisintawee, Ammarin Thakkinstian.

**Data curation:** Auttakiat Karnjanapiboonwong, Usa Chaikledkaew.

**Formal analysis:** Auttakiat Karnjanapiboonwong, Usa Chaikledkaew, Charungthai Dejthevaporn, Ammarin Thakkinstian.

**Funding acquisition:** Usa Chaikledkaew.

**Investigation:** Auttakiat Karnjanapiboonwong.

**Methodology:** Auttakiat Karnjanapiboonwong, Usa Chaikledkaew, Thunyarat Anothaisintawee, Naiyana Praditsitthikorn, Ammarin Thakkinstian.

**Project administration:** Usa Chaikledkaew.

**Supervision:** Usa Chaikledkaew, Thunyarat Anothaisintawee, Naiyana Praditsitthikorn, Charungthai Dejthevaporn.

**Validation:** Auttakiat Karnjanapiboonwong, Usa Chaikledkaew, Thunyarat Anothaisintawee, Naiyana Praditsitthikorn, Charungthai Dejthevaporn, Ammarin Thakkinstian.

**Writing – original draft:** Auttakiat Karnjanapiboonwong, Usa Chaikledkaew, Thunyarat Anothaisintawee, Naiyana Praditsitthikorn, Charungthai Dejthevaporn, Ammarin Thakkinstian.

**Writing – review & editing:** Auttakiat Karnjanapiboonwong, Usa Chaikledkaew, Thunyarat Anothaisintawee, Naiyana Praditsitthikorn, Charungthai Dejthevaporn, Ammarin Thakkinstian.

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
