## [Decision Letter · Decision Letter 0]

25 Jan 2024

PONE-D-23-36269Cost-utility analysis of home blood pressure measurement for screening and diagnosis of hypertension through village health volunteer mechanism in ThailandPLOS ONE

Dear Dr. Chaikledkaew,

Thank you for submitting your manuscript to PLOS ONE. After careful consideration, we feel that it has merit but does not fully meet PLOS ONE’s publication criteria as it currently stands. Therefore, we invite you to submit a revised version of the manuscript that addresses the points raised during the review process.

We look forward to receiving your revised manuscript.

Kind regards,

Desire Aime Nshimirimana, MBChB,Msc

Academic Editor

PLOS ONE

Journal Requirements:

3. We noticed you have some minor occurrence of overlapping text with the following previous publication(s) among others, which needs to be addressed:

https://academic.oup.com/ajh/article/28/5/595/2743434?login=false

https://ecronicon.com/ecemr/pdf/prevalence-and-predictors-of-hypertension-type-2-diabetes-mellitus-comorbidity-among-patients-in-wachemo-university-nigest-elleni-mohammed-memorial-referral-hospital-southern-ethiopia.pdf?

In your revision ensure you cite all your sources (including your own works), and quote or rephrase any duplicated text outside the methods section. Further consideration is dependent on these concerns being addressed.

This work was supported by funding from Mahidol University and the International Decision Support Initiative (iDSI) through the doctoral study in Mahidol University Health Technology Assessment (MUHTA) Graduate Program. This work was produced as part of the iDSI (www.idsihealth.org), which supports countries to get the best value for money from health spending. iDSI receives funding support from the Bill & Melinda Gates Foundation and the UK Department for International Development (Grant No. OPP1087363). The funders had no role in the study design, data collection and analysis, decision to publish, or preparation of the manuscript.

5. We note that your Data Availability Statement is currently as follows: All relevant data are within the manuscript and its Supporting Information files.

Additional Editor Comments

1. Introduction

The introduction has a lot of inconsistencies, grammatical errors and typos. i.e line 78 and 79; you wrote mm/Hg and line 91; you wrote mmHg.

The paragraph 76-83 is not important as it discusses the specificity and sensitivity. I would much prefer reading results of other studies done on cost-effectiveness of ABPM or HBPM. At least 3 studies discussing the cost effectiveness of ABPM or HBPM or CBPM is necessary

The aim of this study is confusing “this study aimed to evaluate the cost-effectiveness of HBPM as either primary or secondary screening after receiving CBPM compared with CBPM only, a current practice”. Kindly re-formulate the aim in an easy way to understand for the reader

2.Methodology

Kindly specify the Study design

Kindly separate ethical consideration from cost and elaborate it alone

Table 1. Parameters used in the model after calibration: This table may be put in the supplementary documents

It is not clear how utility was calculated

Line 138-140:

“The HT incidence and prevalence data were retrieved from the cohort of officers working in

the Electricity Generating Authority of Thailand (EGAT) aged 30 -79 years during 2012-2014 and 2017-2019”

I have a serious concern on the source of the data used in this study. Trusted data should come from a national survey or a census. The electricity generating authority of Thailand (EGT) does not represent the Thailand population. We cannot assume that EGT data can be generalized to the rest of the Thailand population. The use of such data must be highly justified!

In addition, it is not clear how the probabilities of death due to these diseases were calculated (Ischemic stroke, hemorrhagic stroke, CAD, …)

It is also not clear how the transitional probabilities and probability to develop HT among WCHT were calculated

Costs calculation:

1.You said you used a societal perspective in the study, direct medical and non-medical costs were included. Yet, in a societal perspective, indirect costs (the loss of income due to absenteeism at work: salary missed when at the hospital or clinic) must be included. And it is not explained how it was calculated. If there were no costs linked to CBPM, kindly explain this

2. Line 159-161: It is not clear at all how the costs of HT screening was estimated based on HT screening related activities by interviewing health officers and VHV in Chachoengsao province during April-December 2019

Table1: Should be the supplementary documents

Statistical analysis is missing: There should be a paragraph to give details of how statistical analysis was performed including the details of cost-effectiveness plane. I will recommend to pull the cost-effectiveness plane to the main document and explain which procedures are cost-effective

Findings

The results are not presented in the standard manner according to Consolidated Health Economic Evaluation Reporting Standards (CHEERS). Kindly follow the details of CHEERS below;

https://www.valueinhealthjournal.com/article/S1098-3015(13)00022-3/fulltext?_returnURL=https%3A%2F%2Flinkinghub.elsevier.com%2Fretrieve%2Fpii%2FS1098301513000223%3Fshowall%3Dtrue

http://www.ispor.org/TaskForces/EconomicPubGuidelines.asp

1. Study parameters: Reports, ranges, references,--- are missing

2. Incremental costs and outcomes: “For each intervention, report mean values for the main categories of estimated costs and outcomes of interest, as well as mean differences between the comparator groups. If applicable, report incremental cost-effectiveness ratios”

3. Characterizing uncertainty: “Single study-based economic evaluation: Describe the effects of sampling uncertainty for the estimated incremental cost and incremental effectiveness parameters, together with the impact of methodological assumptions (such as discount rate, study perspective)”.

4. Characterizing heterogeneity: “If applicable, report differences in costs, outcomes, or cost-effectiveness that can be explained by variations between subgroups of patients with different baseline characteristics or other observed variability in effects that are not reducible by more information”.

Discussion

Line 248: “Our results indicated that at the societal WTP threshold of 160,000 baht per QALY gained,”. The willingness to pay for Thai peoples could have been discussed in the introduction

Conclusion

The conclusion of this study does not support the results of the study

Reviewers' comments:

Reviewer's Responses to Questions

**Comments to the Author**

1. Is the manuscript technically sound, and do the data support the conclusions?

Reviewer #1: Yes

Reviewer #2: Partly

2. Has the statistical analysis been performed appropriately and rigorously? 

Reviewer #1: Yes

Reviewer #2: I Don't Know

3. Have the authors made all data underlying the findings in their manuscript fully available?

Reviewer #1: Yes

Reviewer #2: No

4. Is the manuscript presented in an intelligible fashion and written in standard English?

Reviewer #1: Yes

Reviewer #2: No

5. Review Comments to the Author

Reviewer #1: Overall, this is a well performed analysis and results provide useful information in the context of middle-income countries. However, it is interesting to note that the cost of treatment of "Serial 1" is distinctly low (77,756 Baht) compared to other strategies (around 108-118,000 Baht) which is quite surprising given that one would expect little differences in hypertension treatment regardless of which method is used for detection or diagnosis. Please explain.

Reviewer #2: The authors present a study comparing three different home blood pressure monitoring strategies (1. Home BP monitoring for all - called HBPM, 2. home monitoring for those with high clinic BP - called serial1, 3. home monitoring for those without high clinic BP - called serial2) in comparison to usual care clinic BP measurement in a Thai community setting. The authors have not followed the CHEERS 2022 reporting guidelines for economic evaluations https://www.equator-network.org/reporting-guidelines/cheers/ which means much of the necessary information is missing. There are grammatical errors throughout, and this journal does not provide copyediting. They report underlying data are all available in-text, but the interview data is not available, only the summary statistics in the parameters table. What interview questions were used to elicit these reported costs? Are they reflecting the program running costs? The model validation is not described clearly - only that it was done by experts. I found the naming convention of "serial1" and "serial2" unintuitive to interpret when looking at the results in tables and figures. Consider renaming or adding a footnote to clarify which is HBPM in people with high clinic BP and which is HBPM in people without high clinic BP. Consider using better contrast in the Figures so they are still readable with greyscale/black and white printing. Figure 3 suggests there is substantial overlap in the ICER estimates generated by the PSA - consider how this influences your certainty in interpreting the results and drawing the conclusion.

6. PLOS authors have the option to publish the peer review history of their article (what does this mean?). If published, this will include your full peer review and any attached files.

Reviewer #1: No

Reviewer #2: No

---

## [Author Response · Author response to Decision Letter 0]

28 Mar 2024

March 28, 2024

Dear Editors,

 On behalf of my co-authors, we would like to resubmit the revised manuscript of our study entitled “Cost-utility analysis of home blood pressure measurement for screening and diagnosis of hypertension through village health volunteer mechanism in Thailand” (Submission ID PONE-D-23-36269) for your kind consideration to be published on PLOS ONE.

We would like to thank all editors and reviewers for their helpful comments and suggestions. We feel that the revised paper is much further improved as a consequence of their inputs. The next page is a point-by point form explaining how we have responded to each comment raised by the editors and reviewers. 

We would like to provide financial statement as follows: “This work was supported by funding from Mahidol University and the International Decision Support Initiative (iDSI) through the doctoral study in Mahidol University Health Technology Assessment (MUHTA) Graduate Program. This work was produced as part of the iDSI (www.idsihealth.org), which supports countries to get the best value for money from health spending. iDSI receives funding support from the Bill & Melinda Gates Foundation and the UK Department for International Development (Grant No. OPP1087363). The funders had no role in the study design, data collection and analysis, decision to publish, or preparation of the manuscript. There was no additional external funding received for this study. "

All authors declare no competing financial interest. We confirm that the present manuscript is original, not previously published, and not submitted for publication or consideration elsewhere. We also anticipate that you will agree with us on the suitability of this manuscript for publication in PLOS ONE. 

Should you have any question, please kindly contact me at usa.chi@mahidol.ac.th. Thank you very much for your kind consideration on this manuscript.

Sincerely yours,

Assoc. Prof. Usa Chaikledkaew, Ph.D.

Corresponding author 

Social Administrative Pharmacy Division, Department of Pharmacy and Mahidol University Health Technology Assessment (MUHTA) Graduate Program, Mahidol University

447 Sri-Ayuthaya Road, Rajathevi, Bangkok 10400, Thailand

Tel: 662-644-8679 ext 5317; Fax: 662-644-8694; 

Email: usa.chi@mahidol.ac.th

Journal Requirements:

Response: Thank you for your suggestion. We ensure that our manuscript meets PLOS ONE’s style requirements, including those for file naming.

Response: Thank you for your suggestion. All data have been reported in our manuscript. 

3. We noticed you have some minor occurrence of overlapping text with the following previous publication(s) among others, which needs to be addressed:

In your revision ensure you cite all your sources (including your own works), and quote or rephrase any duplicated text outside the methods section. Further consideration is dependent on these concerns being addressed.

https://academic.oup.com/ajh/article/28/5/595/2743434?login=false

Response: Thank you for pointing this out. We have paraphrased the sentence and put the citation as follows.

“Thirdly, while prolonged periods of BP measurement may lead to a slight improvement in diagnostic accuracy (52), our study did not fully explore this aspect.”

https://ecronicon.com/ecemr/pdf/prevalence-and-predictors-of-hypertension-type-2-diabetes-mellitus-comorbidity-among-patients-in-wachemo-university-nigest-elleni-mohammed-memorial-referral-hospital-southern-ethiopia.pdf?

Response: We have paraphrased the sentence and put the citation as follows.

“The presence of HT can substantially elevate the risk of various cardiovascular diseases (CVD), including coronary heart diseases (CHD), congestive heart failure (CHF) and peripheral arterial diseases (2). Moreover, it is a major contributor to the incidence of both ischemic and hemorrhagic stroke, as well as renal failure (2).”

This work was supported by funding from Mahidol University and the International Decision Support Initiative (iDSI) through the doctoral study in Mahidol University Health Technology Assessment (MUHTA) Graduate Program. This work was produced as part of the iDSI (www.idsihealth.org), which supports countries to get the best value for money from health spending. iDSI receives funding support from the Bill & Melinda Gates Foundation and the UK Department for International Development (Grant No. OPP1087363). The funders had no role in the study design, data collection and analysis, decision to publish, or preparation of the manuscript. 

Please provide an amended statement that declares all the funding or sources of support (whether external or internal to your organization) received during this study, as detailed online in our guide for authors at http://journals.plos.org/plosone/s/submit-now. Please also include the statement “There was no additional external funding received for this study.” in your updated Funding Statement. Please include your amended Funding Statement within your cover letter. We will change the online submission form on your behalf.

Response: Thank you for your helpful suggestions. We have provided an amended statement that declares all the funding or sources of support (whether external or internal to our organization) received during this study. We have already included the statement “There was no additional external funding received for this study” in our updated Funding Statement within our cover letter. 

5. We note that your Data Availability Statement is currently as follows: All relevant data are within the manuscript and its Supporting Information files.

The values behind the means, standard deviations and other measures reported;

Response: Thank you for your helpful suggestions. Data cannot be shared publicly because there are ethical restrictions on publicly sharing a data set. Data are available from the Research Ethics Protection Unit, the Faculty of Dentistry/Faculty of Pharmacy, Mahidol University (contact via Dental Simulation and Research Building, 5th Floor Mahidol University Faculty of Dentistry, 6, Yothi Road, Ratchathewi, Bangkok, 10400 or +662-200-7622) for researchers who meet the criteria for access to confidential data.

Additional Editor Comments

1. Introduction

The introduction has a lot of inconsistencies, grammatical errors and typos. i.e line 78 and 79; you wrote mm/Hg and line 91; you wrote mmHg.

Response: Thank you for your helpful suggestion. We have already corrected the grammatical errors and typos in the introduction section as suggested.

The paragraph 76-83 is not important as it discusses the specificity and sensitivity. I would much prefer reading results of other studies done on cost-effectiveness of ABPM or HBPM. At least 3 studies discussing the cost effectiveness of ABPM or HBPM or CBPM is necessary

Response: Thank you for your great suggestion. As suggested, we have already revised the sentences as follows (Introduction, page 4, line 78-86). 

“According to a recent systematic review by Wang et al. 2003 (22), four studies compared HBPM to CBPM. Two of these studies focused on HT management (23, 24), while the other two were for diagnosis purpose (25, 26) However, the latter two studies were cost analysis studies that demonstrated medical cost savings in screening with HBPM. Furthermore, the cost-effectiveness study by Lovibond et al. compared screening with ABPM and HBPM to CBPM. The study concluded that screening with ABPM was the most cost-effective strategy across all age groups (50, 60, 70, and 75 years) (27). Additionally, in younger age groups, ABPM yielded greater cost savings but was associated with a slight reduction in quality adjusted life years (QALYs); nevertheless, it remained the most cost-effective option (27).”

The aim of this study is confusing “this study aimed to evaluate the cost-effectiveness of HBPM as either primary or secondary screening after receiving CBPM compared with CBPM only, a current practice”. Kindly re-formulate the aim in an easy way to understand for the reader

Response: Thank you for your helpful suggestions. We have already revised the aim of this study as follows (Introduction, page 4, line 98-99). 

“Therefore, the objective of this study was to assess the cost-effectiveness of BP screening strategies in Thailand.”

2. Methodology

Kindly specify the Study design

Response: Thank you for your very helpful suggestion. We have already put sub-heading “Study design” in the Materials and Methods section (Study design, page 5, line 103-109). 

“We conducted a cost-utility analysis using a Markov model to compare costs and health outcomes of BP screening strategies during lifetime horizon with one-year cycle length. length. Health outcomes were life years (LYs) and QALYs, the multiplication of LYs and utility score. The analysis was performed based on a societal perspective, considering both future costs and health outcomes, which were discounted at a rate of 3% per year as recommended by the Thai HTA guidelines (31).”

Kindly separate ethical consideration from cost and elaborate it alone

Response: Thank you for your very useful suggestions. We have separated ethical consideration from cost and elaborate it alone under sub-heading “Ethics approval” (page 5, line 114-117) as below. 

“The ethical approval of this study was granted by the Faculty of Dentistry/Faculty of Pharmacy, Mahidol University. The written informed consent was obtained from the VHV and local officers who participated in the interviews for data collection on costs.”

Table 1. Parameters used in the model after calibration: This table may be put in the supplementary documents

Response: Thank you for your helpful suggestion. We have already put Table 1 in the supporting information as “S1 Table. Parameters Used in the Model after Calibration”.

It is not clear how utility was calculated

Line 138-140:

Response: Thank you for raising this point. We have added the sentence to explain how utility was calculated under “Utility” sub-heading (page 9, line 207-209).

“The proportion of stroke cases in all CVD cases was 0.64, while the proportion of CAD in all CVD was 0.36 (47). Therefore, the overall utility of CVD was calculated as (0.55 x 0.64) + (0.75 x 0.36) = 0.62.”

“The HT incidence and prevalence data were retrieved from the cohort of officers working in

the Electricity Generating Authority of Thailand (EGAT) aged 30 -79 years during 2012-2014 and 2017-2019”

I have a serious concern on the source of the data used in this study. Trusted data should come from a national survey or a census. The electricity generating authority of Thailand (EGT) does not represent the Thailand population. We cannot assume that EGT data can be generalized to the rest of the Thailand population. The use of such data must be highly justified!

Response: Thank you for raising this point. We have added more explanations on EGAT study under Model input parameter sub-heading as follows (page 7, line 150-155). 

“The HT incidence and prevalence data were obtained from the cohort of 2,235 officers aged 30 -79 years employed by the Electricity Generating Authority of Thailand (EGAT) during two periods: 2012-2014 and 2017-2019. The EGAT study represents one of the largest longitudinal cardiovascular cohort studies in the Thai population. Participants with a wide range of socio-demographic backgrounds located in Bangkok and three different sites in Western and Northern Thailand were randomly enrolled in this study.”

In addition, we also have added the sentences in Discussion part as one of the limitations in this study as follows (page 17, line 379-386).

“Lastly, our study utilized data from the EGAT cohort, which is one of the largest longitudinal cardiovascular cohort studies in the Thai population. The HT incidence estimated from this cohort might not perfectly represent the entire country, as the socio-economic status of EGAT employees differed from that of many economically disadvantaged individuals in Thailand. Additionally, the ‘healthy worker effect’ might influence baseline BP profiles from the EGAT cohort, further limiting the generalizability of our results (55). Nevertheless, the data from EGAT cohort provide the best available evidence for Thai population with HT.”

In addition, it is not clear how the probabilities of death due to these diseases were calculated (Ischemic stroke, hemorrhagic stroke, CAD, …)

Response: Thank you for your raising this point. The probabilities of death due to hemorrhagic stroke, ischemic stroke, and CAD were obtained from published literature reviews. We have added the sentence as follows (page 7, line 159-162). 

“In this model, CVD consisted of hemorrhagic stroke, ischemic stroke, and CAD. The prognosis of CVD varied depending on the specific disease and could be classified as acute or chronic. The probabilities of death due to hemorrhagic stroke, ischemic stroke, and CAD were borrowed from published literature reviews (32-35).”

It is also not clear how the transitional probabilities and probability to develop HT among WCHT were calculated

Response: Thank you for pointing this out. We have added the sentence to explain how the probability to develop HT among WCHT individuals was calculated as follows (page 8, line 172-174). 

“The probability of developing HT among individuals with WCHT was assumed to be equal to the incidence of HT among the general population.”

Costs calculation:

1. You said you used a societal perspective in the study, direct medical and non-medical costs were included. Yet, in a societal perspective, indirect costs (the loss of income due to absenteeism at work: salary missed when at the hospital or clinic) must be included. And it is not explained how it was 

---

## [Decision Letter · Decision Letter 1]

1 Aug 2024

Cost-utility analysis of home blood pressure measurement for screening and diagnosis of hypertension through village health volunteer mechanism in Thailand

PONE-D-23-36269R1

Dear Dr. Usa,

We’re pleased to inform you that your manuscript has been judged scientifically suitable for publication and will be formally accepted for publication once it meets all outstanding technical requirements. After careful review of responses to reviewers and editors, your responses were satisfactory and the paper met our publication criteria. 

1. The study presents the results of original research

2. Results reported have not been published elsewhere

3. Statistical analyses were performed to a high technical standard and described in a sufficient detail

4. Conclusions are presented in an appropriate fashion and are supported by the data

5. The article is presented in an intelligible fashion and is written in standard English 

6. The research meets all applicable standards for the ethics of experimentation and research integrity 

7. The article adheres to appropriate reporting guidelines and community standards for data availability 

Kind regards,

Desire Aime Nshimirimana, MBChB,Msc

Academic Editor

PLOS ONE

Additional Editor Comments (optional):

Reviewers' comments:

Reviewer's Responses to Questions

**Comments to the Author**

1. If the authors have adequately addressed your comments raised in a previous round of review and you feel that this manuscript is now acceptable for publication, you may indicate that here to bypass the “Comments to the Author” section, enter your conflict of interest statement in the “Confidential to Editor” section, and submit your "Accept" recommendation.

Reviewer #1: All comments have been addressed

Reviewer #3: All comments have been addressed

Reviewer #4: All comments have been addressed

2. Is the manuscript technically sound, and do the data support the conclusions?

Reviewer #1: Yes

Reviewer #3: Yes

Reviewer #4: Yes

3. Has the statistical analysis been performed appropriately and rigorously? 

Reviewer #1: Yes

Reviewer #3: Yes

Reviewer #4: Yes

4. Have the authors made all data underlying the findings in their manuscript fully available?

Reviewer #1: Yes

Reviewer #3: Yes

Reviewer #4: Yes

5. Is the manuscript presented in an intelligible fashion and written in standard English?

Reviewer #1: Yes

Reviewer #3: Yes

Reviewer #4: Yes

6. Review Comments to the Author

Reviewer #1: Overall, the authors have adequately addressed all issues. However, the format of some references look odd. Please follow the journal format on the reference style, especially those the sources that are not from international journals. Please make sure all references are formatted appropriately.

Reviewer #3: This study examines the cost-utility analysis of home blood pressure (BP) measurement for screening and diagnosis of hypertension (HT) through village health volunteer mechanism in Thailand. The manuscript contributes valuable insights into the ongoing debate on the economic and clinical viability of home blood pressure measurement for screening and diagnosis of hypertension. The findings from this study could serve as evidence informed policy information for policymakers in determining which BP screening strategy should be implemented in Thailand. However, the manuscript requires some modifications or clarifications and potential structural changes to the model before it can be published. Detailed comments on each section will be provided later in the review.

Introduction

#1. I would recommend the author give more information about these four blood pressure screening strategy. In this way, the importance and necessity of cost-effectiveness can be emphasized.

Method

#2. The methods section is confusing, with several paragraphs or strings of sentences requiring re-wording so that the reader can grasp what has been done. In general, the model to me is more like a HT diagnosis study than screening. For screening, the model starts based on population prevalence data of this disease, through the screening (according to sensitivity and specificity) to identify individuals to correct health states, and then let them come into the natural history of the disease. Maybe the authors need to add more clarifications.

#3. Include a brief rationale for selecting a lifetime horizon.

recommendation for continued breast imaging examinations following treatment. We also added the statements above in the section 2.2 in our revised manuscript. Thank you.

#4. The author mentioned the present study used a societal perspective but there were no indirect costs listed in Table 1.

Results

#5. The results of a lifetime horizon simulation were presented, please add some explanation in the methods for why conducting three model (HPBM vs. CPBM; Serial1 vs. CPBM; Serial2 vs. CPBM).

#6. Putting in the ICERs for each model in the Table 1 would be clear for readers.

#7. Please provide the Figures for conducting the one-way and probabilistic sensitivity analyses.

Discussion

#8. Please also add the strengths or significances of the present study in the Discussion section.

Reviewer #4: This study is well-conducted and well-written. All comments have been appropriately addressed. It is acceptable to published in this current form.

7. PLOS authors have the option to publish the peer review history of their article (what does this mean?). If published, this will include your full peer review and any attached files.

Reviewer #1: No

Reviewer #3: No

Reviewer #4: No

---

## [Editor Report · Acceptance letter]

10 Sep 2024

PONE-D-23-36269R1 

PLOS ONE

Dear Dr. Chaikledkaew, 

I'm pleased to inform you that your manuscript has been deemed suitable for publication in PLOS ONE. Congratulations! Your manuscript is now being handed over to our production team.

Kind regards, 

on behalf of

Dr Desire Aime Nshimirimana 

Academic Editor

PLOS ONE